# Presence of Depression Is Associated with Functional Impairment in Middle-Aged and Elderly Chinese Adults with Vascular Disease/Diabetes Mellitus—A Cross-Sectional Study

**DOI:** 10.3390/ijerph20021602

**Published:** 2023-01-16

**Authors:** Yuxiao Zhao, Yueying Zhang, Kayla M. Teopiz, Leanna M. W. Lui, Roger S. McIntyre, Bing Cao

**Affiliations:** 1Key Laboratory of Cognition and Personality, Faculty of Psychology, Ministry of Education, Southwest University, Chongqing 400715, China; 2Mood Disorders Psychopharmacology Unit, Toronto, ON M5T 2S8, Canada; 3Department of Psychiatry, University of Toronto, Toronto, ON M5T 1R8, Canada; 4Department of Pharmacology, University of Toronto, Toronto, ON M5S 1A8, Canada; 5National Demonstration Center for Experimental Psychology Education, Southwest University, Chongqing 400715, China

**Keywords:** depression, vascular diseases, functional disability, older adults, diabetes

## Abstract

Objectives: The association between chronic diseases and depression has received increasing attention, and are both considered to increase the risk of functional impairment. However, previous research evidence is controversial. Our study aimed to investigate the association between depression, three types of vascular disease (i.e., hypertension, myocardial infarction, stroke), diabetes mellitus, and functional impairment in middle-aged and elderly Chinese people. Methods: We designed a cross sectional study. Data were collected from the China Health and Retirement Longitudinal Study (CHARLS) in 2018. Logistic regression models were used to explore the association between independent variables and functional status. Results: Lower functional status was significantly associated with comorbid depression and vascular disease/diabetes mellitus (Activity of Daily Living/Instrumental Activity of Daily Living: Adjusted OR of Hypertension, Diabetes mellitus, Myocardial infarction, Stroke is 3.86/4.30, 3.80/4.38, 3.60/4.14, 6.62/7.72, respectively; all *p* < 0.001). Conclusions: Depression is associated with functional decline in middle-aged and elderly Chinese individuals with vascular disease/diabetes mellitus. Identifying mediational factors and preventative strategies to reduce concurrent depression in persons with vascular diseases should be a priority therapeutic vista.

## 1. Introduction

Depression has become the largest contributor to burden of disease and non-fatal health loss due to its high prevalence, chronicity, and comorbidity with physical illness [1,2]. Depression is characterized by a state of low mood and loss of interest, as well as additional symptoms including but not limited to insomnia, fatigue, and difficulty to concentrate [3]. Additionally, vascular disease and diabetes mellitus have been reported to significantly impact quality of life quality in elderly persons (i.e., persons > 65 years old) due to chronicity and poor prognosis [4].

Extant research has reported that depression is strongly associated with vascular disease (i.e., hypertension, myocardial infarction, stroke), as well as diabetes mellitus [5]. The vascular depression hypothesis posits that the phenomenology of depression may be subserved by cardiovascular abnormalities [6]. It has hypothesized that vascular diseases may result in cerebral ischemia in select neurophysiological circuits that are relevant to depressive symptoms, which may partially explain the relationship between vascular diseases and depression [7]. A separate line of evidence suggests that depression may increase the risk for vascular pathologies [8,9]. Multiple mechanisms mediating the preceding relationship include but are not limited to, cortisol dysregulation, arrhythmias, catecholamine signaling, and inflammation [7]. However, the relationship between depression and vascular disease has not been fully characterized, and requires further investigation.

Extant evidence indicates that depression and vascular disease may both lead to functional impairment; the foregoing observation is especially pronounced in middle-aged (i.e., persons aged 45 to 60) [10] and elderly adults who may experience concurrent, natural age-related functional decline [11,12]. Depression is a highly debilitating condition associated with significant functional impairment across multiple domains of psychosocial function, such as difficulty in Activity of Daily Living (ADL) and Instrumental Activity of Daily Living (IADL) [13,14]. It is separately reported that individuals who diagnosed with vascular diseases (e.g., stroke) exhibit greater susceptibility to dementia and cognitive decline due to potential neuropathological changes and vascular lesions in the brain [15,16,17]. Moreover, vascular diseases such as cardiovascular diseases are associated with subsequent functional disability in middle-aged and elderly people [18,19,20,21]. The mechanisms that mediate the foregoing observations are not fully understood.

Notwithstanding, there is insufficient characterization of the potential combined effects of comorbid depression and vascular diseases on measures of functional impairment. Moreover, the effect of depression and comorbid vascular disease/diabetes mellitus on functional impairment in elderly adults has not been fully investigated. There is a need to elucidate the foregoing relationship among individuals with diabetes mellitus given that a substantial proportion of individuals with vascular disease have a comorbid diagnosis of diabetes mellitus [4].

Herein, considering comorbidity with chronic diseases and depression has always been researching focuses, we adopted a cross-sectional design to investigate the associations between self-reported depression and three types of vascular diseases (i.e., hypertension, myocardial infarction and stroke) as well as diabetes mellitus, and the associations between depression and vascular diseases/diabetes mellitus and functional disability, by using a large sample of Chinese middle-aged and elderly population. According to “vascular depression” hypothesis [6], we hypothesized that individuals with comorbid depression and vascular diseases/diabetes may experience worsening of symptoms in both conditions, which may mediate increased functional impairment. And those who have one of these two pathologies may have greater likelihood to develop the other.

## 2. Methods

### 2.1. Subjects

Data were drawn from the fourth wave (2018) survey of China Health and Retirement Longitudinal Study (CHARLS) dataset. The CHARLS dataset is an ongoing longitudinal study led by the National School of Development in Peking University. This dataset includes middle-aged and elderly adult participants aged 45 years and older using probability proportional to size (PPS) sampling in 150 counties, 450 communities/villages from 28 provinces of mainland China [22]. The surveys were conducted every 2 to 3 years, with a total of four waves from 2011 to 2018. Data were be excluded if participants died during the year selecting data. All participants were interviewed in-person at their homes. The fourth-wave survey was used in this analysis, in which we merged the Demographic Background dataset and Health Status & Functioning dataset of subjects. The inclusion criterion was: the participant is aged at least 45 years old. Participants who have missing values in self-reported depressive score or all four types of chronic diseases, and all kinds of functional disabilities were excluded.

The CHARLS was approved by the Institutional Review Board of Peking University granted ethical consent (IRB00001052-11015). Written informed consent was obtained from each participant.

### 2.2. Measures

#### 2.2.1. Vascular Diseases/Diabetes Mellitus and Depression

Participants were asked to report whether they had a diagnosis of vascular disease or diabetes mellitus obtained by a physician. Participants were also asked to specify the type of disease diagnosis (e.g., “have you been diagnosed with hypertension by your physician?”). All interviews were conducted by trained investigators [22].

Depressive symptoms were evaluated using the 10-item Center for the Epidemiological Studies of Depression Short Form (CES-D-10), a widely used measure for screening depressive symptoms in primary care settings [23] Of the 10-items on the CES-D-10, 8 items assess the frequency of negative feelings or behaviors (e.g., “I had trouble keeping my mind on what I was doing”). Responses are rated on a 4-point scale from 1 (less than one day) to 4 (5–7 days). Another two items (i.e., item 5 and item 8) ask about the frequency of positive feelings or behaviors (e.g., “I felt hopeful about the future”). Responses for the foregoing two items are reverse rated (i.e., selecting “less than one day” receives a score of 3, and selecting “5–7 days” receives a score of 0). The total CES-D-10 score ranges from 0 to 30, with a higher score indicating a higher level of depressive symptoms. A cutoff score of ≥10 was used for depression classification.

#### 2.2.2. Functional Impairment

The Activities of Daily Living (ADL) scale and Instrumental Activities of Daily Living (IADL) scale were used to assess participants’ functional disability. The ADL scale contains 6 items which asked about whether participants have difficulty in six daily activities (i.e., dressing, bathing, feeding, transferring, toileting and continence) [24]. The IADL scale contains 6 items focusing on six instrumental daily activities (i.e., household chores, preparing hot meals, shopping for groceries, making phone calls, taking medications and managing money) [25]. There are four options in each item for both sales: (1) No, I don’t have any difficulty; (2) I have difficulty but can still do it; (3) Yes, I have difficulty and need help; (4) I cannot do it. Participants classified as independent would need to complete all the activities without difficulty. However, participants would be classified as dependent if they reported to have had difficulty in completing any of the foregoing activities [22].

#### 2.2.3. Covariates

Multiple covariates were incorporated in our statistical analysis, including age, gender, education level (i.e., coded into three levels of education: 1-below primary, 2-primary and junior school, 3-high school and above), marital status (i.e., divided into two categories: married/cohabitating, divorced/separated/widowed/never married), drinking (i.e., drink more than once a month/less than once a month/do not drink) and smoking (never smoke/quit smoke/smoke).

### 2.3. Statistical Analysis

Descriptive statistics was performed to summarize sociodemographic characteristics. Categorical variables were summarized with frequencies and proportions. Continuous variables were provided with means and standard deviations (SDs). An independent-sample *t*-test and chi-square test were performed to explore whether there is a difference in depressive scores and depression rate according to the diagnosis of vascular diseases and diabetes mellitus.

The association between various groups (categorized by depression & all types of vascular disease) and ADL, IADL-related disability was tested using the logistic regression models. The unexposed group was taken as a reference category, and the odds ratio (OR) of being dependent (i.e., functionally disabled) versus being independent in each group compared to unexposed groups was calculated. Both raw and adjusted ORs as well as their 95% confidence intervals (Cis) were reported. Two-tailed *p*-values below 0.05 were considered statistically significant.

## 3. Results

### 3.1. Sociodemographic Characteristics of Subjects

A total of 10531 participants were included in the analysis herein (Figure 1). Table 1 presents the baseline characteristics of all subjects. Excluding individuals who had missing data regarding diagnosis, the prevalence rates of hypertension, diabetes mellitus, myocardial infarction, and stroke were 14.6%, 5.8%, 7.9%, and 4.5%, respectively, and the participants who without all four diseases accounts for 42.9%. The average ages of patients with the five foregoing groups were 61.8 (SD = 9.4), 62.0 (SD = 8.8), 62.4 (SD = 9.0), 65.1 (SD = 8.2), 60.5 (SD = 9.3) years old, respectively. The prevalence of depression across each group was 42.0%, 42.4%, 46.5% and 50.1%, 42.0%, respectively.

### 3.2. The Association between Depression and Vascular Diseases/Diabetes Mellitus

A comparison of depressive scores according to whether participants had a codified diagnosis of vascular diseases is shown in Table 2. Depressive severity scores were significantly higher in participants who had vascular diseases/diabetes mellitus when compared to participants without the foregoing diseases (*p* < 0.001). Chi-square test results indicated that the prevalence rate of depression was also significantly higher in participants who were diagnosed with any type of the chronic diseases (i.e., Hypertension: OR = 1.39, 95% CI: 1.25, 1.54; Diabetes mellitus: OR = 1.32, 95% CI: 1.14, 1.52; Myocardial infarction: OR = 1.67, 95% CI: 1.48, 1.90; Stroke: OR = 1.80, 95% CI: 1.54, 2.10, Unexposed group: OR = 1.32, 95% CI: 1.18, 1.48, separately; all *p* < 0.0001) (Table 2).

### 3.3. Functional Impairment According to Groups

Table 3 shows descriptive statistics about functional ability as measured by the ADL scale and IADL scale. Table 3 highlights the number of “functionally independent” people as well as the proportion of “functionally independent” people reported in each disease and unexposed category.

The association between ADL, IADL-related disability, and depression or vascular diseases/diabetes mellitus is separately presented in Table 4, Table 5 and Table 6. On the whole, the diseased group had higher risk of functional impairment compared to not having diseases group (Table 4). Stroke patients had higher Ors in both ADL and IADL-dependent assessments compared to unexposed participants; however, the foregoing associations were not observed in the diabetes mellitus subgroup. We also observed that myocardial infarction patients had higher Ors in ADL-dependent assessments compared to the unexposed group. Moreover, hypertension patients had higher Ors in IADL-dependent assessment. Furthermore, a strong association between depression and functional disability was identified; the foregoing association remained significant after controlling for extraneous variables including age, gender, education level, marital status and health behaviors (i.e., drinking and smoking). Detailed results are shown in Table 5 and Table 6.

Additionally, we observed that subjects who reported comorbid depression and vascular disease had significantly higher Ors with respect to higher scores of functional impairment in compared to those unexposed, and the association still remained after adjusted. (ADL: Hypertension: AOR = 3.863, 95% CI: 3.028, 4.927; Diabetes mellitus: AOR = 3.803, 95% CI: 2.844, 5.084; Myocardial infarction: AOR = 3.604, 95% CI: 2.787, 4.660; Stroke: AOR = 6.615, 95% CI: 4.914, 8.905, respectively; all *p* < 0.001; IADL: Hypertension: AOR = 4.295, 95% CI: 3.552, 5.194; Diabetes mellitus: AOR = 4.380, 95% CI: 3.441, 5.576; Myocardial infarction: AOR = 4.144, 95% CI: 3.367, 5.100; Stoke: AOR = 7.718, 95% CI: 6.003, 9.922, respectively; all *p* < 0.001).

## 4. Discussion

Herein, we explored the association between self-reported depression as well as vascular diseases (i.e., hypertension, myocardial infarction, and stroke)/diabetes mellitus and functional outcomes in middle-aged and older Chinese adults. We evaluated functional impairment as measured by the ADL scale and IADL scale and the fourth wave (2018) dataset from the CHARLS. Taken together, we observed greater functional impairment in subjects who had concurrent depression when compared with those unexposed.

Our findings are in accordance with published literature evaluating the effect of associations between comorbid medical disorders and mental disorders on functional outcomes. Extant literature has reported on the deleterious impact of vascular diseases/diabetes mellitus on cognition and functional abilities [26]. For example, a previous meta-analysis revealed a significant increase in physical disability in individuals with diabetes [27]. Moreover, it has been separately reported that individuals with hypertension, myocardial infarction, or stroke have an increased risk of functional impairment [19,20]. Furthermore, a separate retrospective cohort study reported on the increased risk of dementia in patients with comorbid depression and vascular disorders compared to individuals with a sole diagnosis of depression, especially for patients with stroke or hypertension [15].

Available evidence indicates that elevated levels of C-reactive protein and albuminuria in people with cardiovascular diseases (CVD), which are biomarkers of CVD (e.g., myocardial infarction, stroke, coronary heart disease etc.), were independently correlated with functional disability in older adults [28]. The preceding finding implies a close association of inflammation and albuminuria in persons with CVD and functional impairment. However, the mechanisms subserving the observed associations require further research.

It is also well described that depression adversely affects ADLs, with severe impairment in ADLs noted in geriatric populations [29]. Previous evidence indicates that individuals with depression perform worse across disparate neuropsychological measures (e.g., executive function) when compared with non-depressed persons [30,31]. In our analysis, we did not identify a main effect of all the vascular diseases on functional impairment. It is likely that stroke is CVD with sudden onset and typically lead to direct damage to functional abilities, the extent of harm of diabetes depends more on the effect of covariates (e.g., duration of the disease, gender) [21]. However, we found that the presence of depression had an association with functional impact of the mentioned above chronic disease.

Previous findings have implicated medical illnesses (e.g., CVD) with the onset of depression (i.e., “vascular depression” hypothesis) [32,33]. Mechanisms subserving the “vascular depression” hypothesis also have been published elsewhere [34]. Among multiple factors reported, vascular diseases are reported to predict the increase of white matter hyperintensities (WMH) associated with depression, indicating a pathophysiologic nexus linking vascular and white matter pathology [7]. The significant association between self-reported depression and vascular diseases/diabetes from our study was in accordance with the “vascular depression” hypothesis. Our findings herein highlight the importance of developing practical treatments for individuals with both depression and vascular diseases/diabetes.

### Limitations

Several limitations need to be considered when interpreting the results of our study. First, the cross-sectional design precludes any statements about causation and cannot indicate a prediction of future events. Secondly, the measures of functional disability administered in our study were limited. Future studies should include a comprehensive multi-dimensional battery of functional ability assessments. Nonetheless, our study endeavors to extend knowledge further by documenting the additive effects of vascular disease in persons with depression on functional outcomes in elderly Chinese populations. The aging Chinese population along with rising rates of cardiovascular and metabolic disorders across China further underscores the relevance of our findings.

Taken together, future research should aim to implement a longitudinal design to explore causal relationships between depression, vascular disease/diabetes mellitus, and functional impairment. Furthermore, the biophysiological factors and mechanism underlying the foregoing associations need to be characterized in order to inform treatment approaches that endeavor to improve functional disability in individuals at risk.

## 5. Conclusions

In summary, our study indicated that depression is associated with functional impairment in middle-aged and elderly Chinese adults with vascular disease/diabetes mellitus. Predicting and preventing functional impairment in persons with depression is a priority therapeutic vista. Parsing underlying mechanisms that subserve the foregoing comorbidities and their effect on poor functional outcomes is a priority research question.

## Figures and Tables

**Figure 1 ijerph-20-01602-f001:**
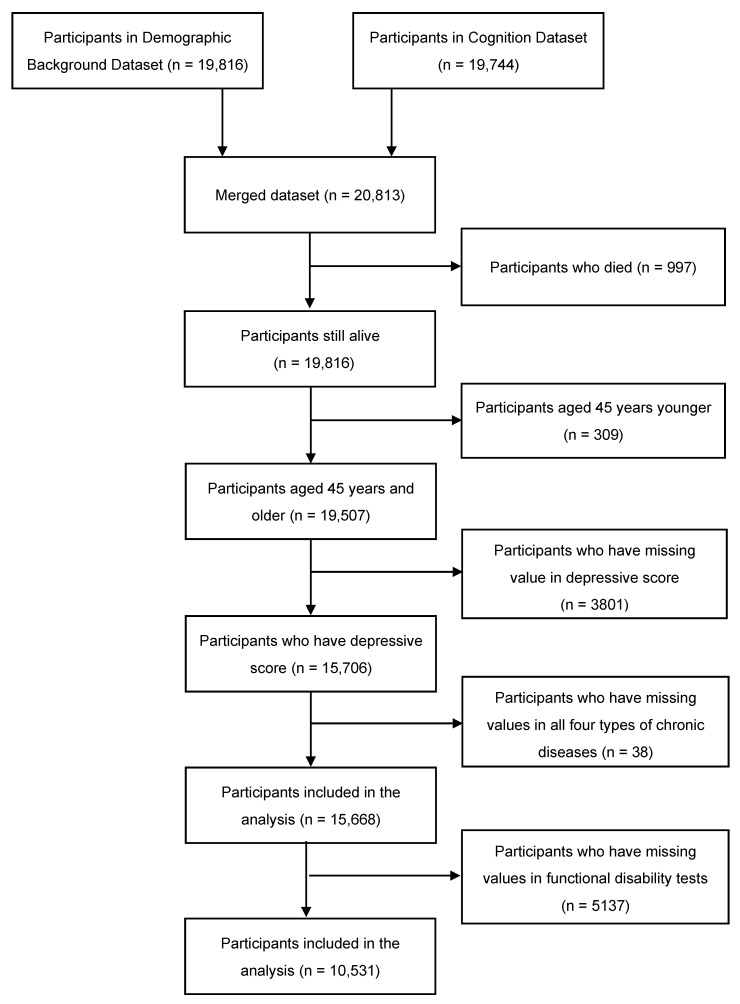
Inclusion process of the participants.

**Table 1 ijerph-20-01602-t001:** Sociodemographic characteristics according to vascular disease status.

Characteristics	Total	Hypertension	Diabetes Mellitus	Myocardial Infarction	Stroke	Unexposed **
**Total (*n*, %)**	10,531	1692 (14.6)	834 (5.8)	1082 (7.9)	691 (4.5)	4519 (42.9)
**Age** (years; Mean ± SD)		61.8 ± 9.4	62.0 ± 8.8	62.4 ± 9.0	65.1 ± 8.2	60.5 ± 9.3
**Gender (*n*, %)**						
Male		899 (53.1)	395 (47.4)	464 (42.9)	360 (52.1)	1917 (42.4)
Female		793 (46.9)	439 (52.6)	618 (57.1)	331 (47.9)	2602 (57.6)
**Education level (*n*, %)**						
Illiterate		315 (18.6)	172 (20.6)	215 (19.9)	147 (21.3)	998 (22.1)
Primary school and below		752 (44.4)	354 (42.5)	453 (41.9)	302 (43.7)	2132 (47.2)
Junior high school or above		615 (37.0)	308 (36.9)	414 (38.2)	242 (35.0)	1389 (30.7)
**Mariral status (*n*, %)**						
Married/cohabitating		1458 (86.2)	733 (87.9)	937 (86.6)	566 (81.9)	3959 (87.6)
Divorced/separated/widowed/never married		234 (13.8)	101 (12.1)	145 (13.4)	125 (18.1)	560 (12.4)
**Drinking * (*n*, %)**						
Drink more than once a month		480 (28.4)	190 (22.8)	232 (21.5)	131 (19.0)	1146 (25.4)
Drink less than once a month		1212 (71.6)	644 (77.2)	850 (78.5)	560 (81)	3373 (74.6)
**Smoking (*n*, %)**						
Yes		489 (28.9)	196 (23.5)	242 (22.4)	158 (22.9)	1219 (27.0)
No		898 (53.1)	508 (60.9)	664 (61.4)	373 (54.0)	2787 (61.7)
Quit		305 (18.0)	130 (15.6)	176 (16.2)	160 (23.1)	513 (11.4)
**Depression (*n*, %)**						
Yes		710 (42.0)	354 (42.4)	503 (46.5)	346 (50.1)	1897 (42.0)
No		982 (58.0)	480 (57.6)	579 (53.5)	345 (49.9)	2622 (58.0)

* Drinking: Reference to alcohol. ** Unexposed: Participants without all four diseases.

**Table 2 ijerph-20-01602-t002:** Scores and odds ratio of depression in the presence of vascular diseases.

Vascular Diseases	Depressive Scores (Mean ± SD)			OR * (95% CI)
	Have Diseases	Not Have Diseases	*t*	
**Hypertension**	9.17 ± 0.17	8.03 ± 0.06	6.75 ***	1.39 (1.25, 1.54) ***
**Diabetes mellitus**	9.56 ± 0.25	8.29 ± 0.05	5.5 ***	1.32 (1.14, 1.52) ***
**Myocardial infarction**	10.04 ± 0.21	8.01 ± 0.06	10.09 ***	1.67 (1.48, 1.90) ***
**Stroke**	10.87 ± 0.27	8.28 ± 0.05	10.3 ***	1.80 (1.54, 2.10) ***
**Unexposed**	10.32 ± 0.17	9.16 ± 0.01	5.89 ***	1.32 (1.18, 1.48) ***

* The “Not have disease” group was taken as a reference category, and the odds ratio (OR) of depression’s prevalence rate in “Have diseases” group to “Not have disease” group was calculated. In unexposed group, “Have diseases” indicate for the participants have any one of vascular disease, “Not have diseases” indicate for the participants who free of four diseases *** *p* < 0.001 (two tailed).

**Table 3 ijerph-20-01602-t003:** Functional ability in the presence of vascular disease.

Functional Assessment (Independent *n*, %)	Total	Hypertension	Diabetes Mellitus	Myocardial Infarction	Stroke	Unexposed
	10,531	*n* = 1256	*n* = 644	*n* = 899	*n* = 590	*n* = 4519
**Activity of daily living (*n*, %)**						
Total independent		924 (73.6)	455 (70.7)	630 (70.1)	344 (58.3)	3738 (82.7)
Dressing		1149 (91.5)	586 (91.0)	827 (92.0)	494 (83.7)	4272 (94.5)
Bathing or showering		1118 (89.0)	573 (89.0)	808 (89.9)	465 (78.8)	4277 (94.6)
Eating		1228 (97.8)	625 (97.0)	873 (97.1)	546 (92.5)	4455 (98.6)
Getting into or out of bed		1143 (91.0)	575 (89.3)	817 (90.9)	495 (83.9)	4289 (94.9)
Using the toilet		1044 (83.1)	531 (82.5)	720 (80.1)	437 (74.1)	4070 (90.1)
Controlling urination and defecation		1186 (94.4)	612 (95.0)	834 (92.8)	527 (89.3)	4371 (96.7)
**Instrumental activities of daily living (*n*, %)**						
Total independent		820 (65.3)	403 (62.6)	581 (64.6)	284 (48.1)	3451 (76.4)
Household chores		1012 (80.6)	500 (77.6)	716 (79.6)	393 (66.6)	4028 (89.1)
Preparing hot meals		1080 (86.0)	547 (84.9)	782 (87.0)	438 (74.2)	4183 (92.6)
Shopping for groceries		1123 (89.4)	571 (88.7)	795 (88.4)	469 (79.5)	4275 (94.6)
Making phone calls		1057 (84.2)	545 (84.6)	761 (84.6)	463 (78.5)	3961 (87.7)
Taking medications		1178 (93.8)	593 (92.1)	829 (92.2)	524 (88.8)	4363 (96.5)
Managing money		1097 (87.3)	545 (84.6)	764 (85.0)	449 (76.1)	4110 (90.9)

**Table 4 ijerph-20-01602-t004:** The risk of functional impairment according to vascular disease—single analysis.

Diseases	UOR (95% CI)	*p*-Value	AOR * (95% CI)	*p*-Value
**Activity of daily living (ADL)**				
Hypertentiosn	1.45 (1.26, 1.67)	**<0.001**	1.37 (1.19, 1.58)	**<0.001**
Diabetes	1.45 (1.22, 1.73)	**<0.001**	1.43 (1.19, 1.72)	**<0.001**
Myocardial infarction	1.62 (1.39, 1.89)	**<0.001**	1.57 (1.35, 1.84)	**<0.001**
Stroke	2.54 (2.14, 3.01)	**<0.001**	2.32 (1.95, 2.77)	**<0.001**
Exposed	1.63 (1.42, 1.86)	**<0.001**	1.55 (1.35, 1.78)	**<0.001**
Depression	2.91 (2.65, 3.20)	**<0.001**	2.95 (2.68, 3.25)	**<0.001**
**Instrumental activity of daily living (IADL)**				
Hypertension	1.48 (1.30, 1.68)	**<0.001**	1.43 (1.25, 1.64)	**<0.001**
Diabetes	1.44 (1.22, 1.70)	**<0.001**	1.43 (1.20, 1.70)	**<0.001**
Myocardial infarction	1.43 (1.24, 1.66)	**<0.001**	1.41 (1.21, 1.64)	**<0.001**
Stroke	2.68 (2.27, 3.17)	**<0.001**	2.57 (2.15, 3.06)	**<0.001**
Exposed	1.58 (1.40, 1.79)	**<0.001**	1.55 (1.37, 1.77)	**<0.001**
Depression	3.10 (2.85, 3.38)	**<0.001**	3.10 (2.83, 3.40)	**0.001**

In each category, the not have disease group was taken as a reference category, and the odds ratio (OR) of functional impairment’s risk have diseases group to not have disease group was calculated. * AOR and 95% CI were calculated by adjusting for the potential confounders including age, gender, education level, marital status, and health behaviors (i.e., drinking and smoking). In each category, UOR: univariate odds ratio; AOR: adjusted odds ratio, REF: reference. The *p*-value in bold emphasized statistical significance.

**Table 5 ijerph-20-01602-t005:** The risk of functional impairment according to depression and comorbid vascular disease (activity of daily living).

Variables	*n*	UOR (95% CI)	*p*-Value	AOR * (95% CI)	*p*-Value
**Depression by hypertension**					
Both	620	4.16 (3.43, 5.04)	**<0.001**	3.86 (3.03, 4.93)	**<0.001**
Depression only	2687	3.01 (2.64, 3.44)	**<0.001**	2.93 (2.47, 3.46)	**<0.001**
Hypertension only	636	1.42 (1.13, 1.80)	**0.003**	1.37 (1.01, 1.85)	**0.042**
Unexposed	3410	REF		REF	
**Depression by diabetes**					
Both	322	4.20 (3.32, 5.32)	**<0.001**	3.80 (2.84, 5.08)	**<0.001**
Depression only	4007	2.87 (2.59, 3.18)	**<0.001**	2.82 (2.46, 3.23)	**<0.001**
Diabetes only	322	1.30 (0.97, 1.75)	0.083	1.30 (0.90, 1.89)	0.169
Unexposed	4945	REF		REF	
**Depression by myocardial infarction**					
Both	459	3.95 (3.21, 4.86)	**<0.001**	3.60 (2.79, 4.66)	**<0.001**
Depression only	3433	2.85 (2.55, 3.19)	**<0.001**	2.78 (2.40, 3.21)	**<0.001**
Myocardial infarction only	440	1.78 (1.40, 2.27)	**<0.001**	1.65 (1.21, 2.25)	**0.001**
Unexposed	4458	REF		REF	
**Depression by stroke**					
Both	317	7.19 (5.69, 9.08)	**<0.001**	6.62 (4.91, 8.91)	**<0.001**
Depression only	4335	2.87 (2.60, 3.17)	**<0.001**	2.80 (2.45, 3.18)	**<0.001**
Stroke only	273	2.32 (1.76, 3.07)	**<0.001**	2.32 (1.61, 3.34)	**<0.001**
Unexposed	5359	REF		REF	

* AOR and 95% CI were calculated by adjusting for the potential confounders including age, gender, education level, marital status, and health behaviors (i.e., drinking and smoking). UOR: univariate odds ratio; AOR: adjusted odds ratio, REF: reference. The *p*-value in bold emphasized statistical significance.

**Table 6 ijerph-20-01602-t006:** The risk of functional impairment according to depression and comorbid vascular disease (instrumental activity of daily living).

Variables	*n*	UOR (95% CI)	*p*-Value	AOR *(95% CI)	*p*-Value
**Depression by hypertension**					
Both	620	4.47 (3.73, 5.36)	**<0.001**	4.30 (3.55, 5.19)	**<0.001**
Depression only	2687	3.17 (2.82, 3.57)	**<0.001**	3.17 (2.80, 3.59)	**<0.001**
Hypertension only	636	1.45 (1.18, 1.78)	**<0.001**	1.38 (1.12, 1.72)	**0.003**
Unexposed	3410	REF		REF	
**Depression by diabetes**					
Both	322	4.50 (3.58, 5.66)	**<0.001**	4.38 (3.44, 5.58)	**<0.001**
Depression only	4007	2.99 (2.72, 3.29)	**<0.001**	2.30 (2.72, 3.31)	**<0.001**
Diabetes only	322	1.25 (0.95, 1.63)	0.108	1.27 (0.96, 1.68)	0.092
Unexposed	4945	REF		REF	
**Depression by myocardial infarction**					
Both	459	4.19 (3.43, 5.10)	**<0.001**	4.14 (3.37, 5.10)	**<0.001**
Depression only	3433	2.91 (2.63, 3.22)	**<0.001**	2.94 (2.64, 3.27)	**<0.001**
Myocardial infarction only	440	1.22 (0.96, 1.55)	0.104	1.20 (0.93, 1.53)	0.155
Unexposed	4458	REF		REF	
**Depression by stroke**					
Both	317	8.16 (6.41, 10.38)	**<0.001**	7.72 (6.00, 9.92)	**<0.001**
Depression only	4335	3.08 (2.81, 3.38)	**<0.001**	3.07 (2.79, 3.38)	**<0.001**
Stroke only	273	2.54 (1.97, 3.28)	**<0.001**	2.36 (1.81, 3.08)	**<0.001**
Unexposed	5359	REF		REF	

* AOR and 95% CI were calculated by adjusting for the potential confounders including age, gender, education level, marital status and health behaviors (i.e., drinking and smoking). UOR: univariate odds ratio; AOR: adjusted odds ratio, REF: reference. The *p*-value in bold emphasized statistical significance.

## Data Availability

The datasets that support the findings of this study are available on reasonable request from the corresponding author, B.C.

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
