# Peer review of "Presence of Depression Is Associated with Functional Impairment in Middle-Aged and Elderly Chinese Adults with Vascular Disease/Diabetes Mellitus—A Cross-Sectional Study"

_ijerph, 2023, doi:10.3390/ijerph20021602_

Round 1
Reviewer 1 Report
This manuscript reports a cross sectional analysis of an observational epidemiological study conducted in China. The authors investigated associations between depressive symptoms and functional independence in participants with cardiovascular disease or diabetes.
Major points :
It is unclear whether or not the authors included those without cardiovascular disease in the analyzed population. The flow chart indicates that they were included, but no data are reported for this group. For example, Table 1 provides no characteristics of participants free of the 4 conditions studied. Similarly, Table 2 does not provide a depression score for participants free of the 4 conditions studied. The authors need to clarify this issue. If those free of the 4 conditions were not analyzed, this represents a serious bias that weakens the validity of the study; a new analysis with them might give a better idea of the respective roles of vascular disease and depression on functional independence.
On several occasions, the authors used terms related to prediction or impact. The study is concerned only with associations and cannot indicate a causal relationship or prediction of future events. This should be changed and explained in the limitations section of the discussion. The title should also be changed and the term "association" is more appropriate than "depression increases ... ".
The type of study should be indicated clearly in Title and in the beginning of Methods: i.e. cross sectional study
The respective association between functional impairment and depression and CV/diabetes should be done by a single analysis. The way the authors splits analysis by subgroups is unclear (Table 4 and Table 5). A single analysis could be more appropraite to authors’ hypothysesis.
Minor pointsTable 1 : To drink less than a month is very similar to "do not drink" : these could be combined. Drinking: reference to alcohol should be indicated.
Table 2: Reference condition for OR calculation should be indicated.
Author Response
We are very grateful that your valuable and insightful comments and suggestions on our manuscript. According to your suggestions, we revised our manuscript. Hope you will find the revision has been significantly improved.Please see the attachment.

Reviewer 2 Report
This is an interesting cross-sectional analysis that examined the association between depression and functional impairment in Chinese adults over 45 years who have diabetes or vascular disease. The analysis was appropriately done and the manuscript is generally well-written although I have some recommendations on how to make their manuscript clearer.
Major comments:
1. The title and abstract do not clearly state/indicate that this is a cross-sectional analysis and readers might be misled especially since the data were taken from a longitudinal study. For example, in the title, replace "increases" with "Is Associated with" or something else. The same goes for the "increases" in the abstract and any other language in the paper that might suggest cause and effect rather than just a statistical association.
2. Abstract: I am not understanding the first sentence. Is this background or a statement of a conclusion?
3. I am just a little confused by figure 1. If this was a cross-sectional study, why exclude those who died?
4. The first sentence needs to be a bolder declaration of the novel findings of the study and why these findings are important to extending the literature on this topic. Currently, it starts by restating the purpose of the study. Is your analysis merely to verify other studies or is your study unique? I would be more likely to rate your study higher if you could be more (legitimately) convincing that your results are novel.
Minor comments:
1. Is it necessary to go to three decimal places for all of these ORs and CIs? It makes your results a bit cluttered.
2. The use of "heart attack" should be replaced with "myocardial infarction" to be less colloquial.
3. Abstract should read "Our study aimed..." to be consistent with the use of past tense.
4. Line 37 delete "etc."
5. Line 62. Replace the subjective "suffered" with the objective "diagnosed with" or "reported having"
6. Line 95 if only one requirement for inclusion it should be "The inclusion criterion was..."
Author Response
We are very grateful that your valuable and insightful comments and suggestions on our manuscript. According to your suggestions, we revised our manuscript. Hope you will find the revision has been significantly improved. Please see the attachment.

Round 2
Reviewer 1 Report
The authors have improved their manuscript. They have provided more details about the methods and the population they studied. Unfortunately, this raises serious methodological pitfalls that invalidate their results. In particular, despite my comment, they still do not provide data on a significant portion of the sample included in the study. The statistical analysis is done and reported incorrectly.
Author Response
We are very grateful that your valuable and insightful comments and suggestions on our manuscript. According to your suggestions in round 1 and round 2, we revised our manuscript again. All comments were addressed and incorporated into the new manuscript. After making the suggested modifications, it is our belief that the manuscript has been substantially improved. We have made a data supplement on the unexposed group in our sample, and added a single analysis to improve our statistical analysis. We also changed the tables and text correspondingly. We hope this revision of the manuscript can be accepted by the journal. Please see the attachment.
